# Stabilization Activity of Kelp Extract in Ethylene–Propylene Rubber as Safe Packaging Material

**DOI:** 10.3390/polym15040977

**Published:** 2023-02-16

**Authors:** Traian Zaharescu

**Affiliations:** 1National Institute for Electrical Engineering, Advanced Research (INCDIE ICPE CA), 313 Splaiul Unirii, 030138 Bucharest, Romania; traian.zaharescu@icpe-ca.ro; 2ROSEAL SA, 5 A Nicolae Bălcescu, Odorheiu Secuiesc, 535600 Harghita, Romania

**Keywords:** ethylene–propylene copolymer, stabilization, irradiation, chemiluminescence

## Abstract

This paper presents the stabilization effects of the solid extract of kelp (*Ascophyllum nodosum*) on an engineering elastomer, ethylene–propylene copolymer (EPR), which may be used as packaging material. Progressive increase in additive loadings (0.5, 1, and 2 phr) increases the oxidation induction time for thermally aged rubber at 190 °C from 10 min to 30 min for pristine material and modified polymer by adding 2 phr protection powder. When the studied polymer is γ-irradiated at 50 and 100 kGy, the onset oxidation temperatures increase as a result of blocking the oxidation reactivity of free radicals. The stabilization effect occurs through the activity of alginic acid, which is one of the main active components associated with alginates. The accelerated degradation caused by γ-exposure advances more slowly when the kelp extract is present. The OOT value for the oxidation of EPR samples increases from 130 °C to 165 °C after the γ-irradiation of pristine and modified (2 phr of kelp powder) EPR, respectively. The altered oxidation state of EPR samples by the action of γ-rays in saline serum is faster in neat polymer than in stabilized material. When the probes are placed in physiological serum and irradiated at 25 kGy, the OOT value for neat EPR (145 °C) is much lower than the homologous value for the polymer samples protected by kelp extract (153 °C for the concentration of 0.5 phr, 166 °C for the concentration of 1 phr, and 185 °C for the concentration of 2 phr).

## 1. Introduction

The presence of an efficient stabilization compound is an essential condition for the safety applications of long-life materials with special reference to packaging materials, which are always subjected to stressing factors such as heat and light. Differences in the protection activities of various antioxidants relate to their efficiencies as well as the manner through which they are produced. Material durability is highly related to structural stability, which depends on the characteristics of raw products and the blending formulation. Though synthesis compounds such as hindered phenols or amines are appropriately used for the preservation of stability under hazardous conditions [1,2], they are converted into quinone structures, which poisonous to the human body, when they act in oxidizing polymers [3]. Attempts to diminish oxidation rate are generally based on the substitution of mobile protons from existing hydroxyl moieties in the stabilizing molecules [4], whose mobility determines whether lifetime is extended. An important factor that mitigates polymer oxidation is the material’s strength against oxidative ageing [5], when the local concentration of free radicals, the intermediates that sustain the propagation of degradation, does not reach a critical amount [6].

The natural antioxidants that result from the application of extraction procedures on herbs and plants have been increasingly used, particularly in cases where their use is compatible with human factors [7]. The chain-breaking action of natural antioxidants operates at various temperatures [8], when fragmentation moieties are subjected to the attack of oxygen that diffuses inside the polymer bulk. Fortunately, they may be associated with other types of stabilizers, with which synergistic effects are obtained [9]. Several interesting applications of these types of stability improvement are reported [10,11,12,13,14], illustrating their largely assumed potential.

The healthy contributions of phytocompounds to the quality of food is generally accepted either for preservation of properties [15] or for packaging materials [16]. The extension of dietary applications of algal extracts covers a large area of interest, where the use of algal powders prevents oxidation through their various classes of compounds: polyphenols, phlorotannins, carotenoids, polysaccharides, and polyunsaturated fatty acids [17]. The most attractive effects of natural antioxidants are revealed by their dietary use for several diseases [18,19,20].

Current investigations into the curative properties of natural extracts have inspired detailed examinations of the antioxidant activities of algal extracts [21]. Based on their active components with outstanding antioxidant features, algal extracts have become a main source of oxidation protectors in various areas: medicine [22], pharmaceutics [23], cosmetics [24], nutritional sources [25], ecological packaging [26], and many others. Starting from convenient production technologies [27], algal extracts are suitable materials for anti-ageing protectors in plastics according to some previous assays [28].

The presence of antioxidants in *Ascophyllum nodosum* as 2605.9 mg GAE/100 g of brown seaweed powder [29] shows the enormous potential of these macroalgae to provide healthy extraction materials for the production of ecological plastics. From the group of brown seaweeds, *Ascophyllum nodosum* has the highest antioxidant content [30], which explains the large interest in medicine, food, and cosmetic industries in this seaweed. The abundance of phenolic compounds (2605.89 ± 192.97 mg gallic acid equivalents/100 g dm) in this algal extract [31] recommends it as a pertinent and attractive stabilizer for many applications, especially where it is compatible with needs for natural bioactive additives.

The wide spread of *Ascophyllum nodosum* in different types of forests [32] and the ease of harvesting these seaweeds [33] indicate commercial availability and ready access [34,35]. The brown seaweeds, including *Ascophyllum nodosum*, contain preponderantly alginic acid and alginates [36] as active elements having antioxidant properties [37]. Active components existing in the extracts act successfully as biostimulators [38] because of their contributions as growing agents and antioxidant preventers.

The alginic acid and alginates indicated in the structure presented in Figure 1 belong to the natural polysaccharide class, which exists in the majority of spice and herb extracts in various proportions. The large variety of alginate structures offers support for the wide scale of antioxidant capacities, where their compositions (mannuronic/guluronic (M/G) ratio) vary [39].

The extension of functional features of alginic acid and alginates on the antioxidant range is characterized by the retardation of peroxidation of lipids [40]. The abundance of stabilization elements such as carotenoids in marine resources [29], bioactive compounds such as catechins [41], and seaweed phenolic antioxidants [42] are significant indications for potential uses of seaweed in anti-ageing products. Brown seaweeds are sources of alginate-based compositions available for protective applications [43,44,45]. The variety of preparation procedures starting from macroalgae [46,47,48,49] is an open door for the usage of alginates as efficient protectors against oxidation in polymer composites [28,50,51]. The stability of alginic acid and alginates as feedstocks for composites is explained by their ability to retain free molecular fragments [52,53,54]. Seaweeds, which are very rich in polysaccharides and antioxidants [55], may be considered as a potential source of stabilizers if the polymers are involved in ecological applications or in health care development.

The destructive action of high energy radiation is described by the superunitary value of radiochemical scission yield (*G*(s) = 1.3) for Na-alginates [56], which describes the degradation trend of this solid type of polysaccharide. This procedure of radiation processing is appropriate for the conversion of numerous polysaccharides into useful products as fertilizers [57], water absorbers [58], hydrogels for drug delivery [59], and high-yield biogas production [60].

However, structures belonging to the polysaccharide class of polymers show the formation of—COOH and—OH functions that are active during the protection of host materials against oxidation [61]. Proof of radiation resistance associated with the preservation of antioxidant capacity of algal extracts under γ-irradiation is the satisfactory protection activity of algal extracts added in polymers [62]. The γ-radiolysis of various alginates produces blends of polyguluronic acid and polymannuronic acid fractions in different mixing proportions, which may be identified by FTIR analysis [63]. These decomposition components are largely used in individual antioxidant protection [63,64], the safe drug delivery of polymers [65], and the crosslinking of alginic acid films [66]. The use of radiolysis mechanism in the processing of alginates [67] shows antioxidant features [68] that support the continuous protection activity in the irradiated EPR samples. This situation is similar to polymer modification by rosemary extracts [69], which is very efficient due to the antioxidant activities of descendants of rosmarinic [70] and carnosic [71] acids. A comparison study on the stabilization efficiency of natural extracts in protecting food packaging materials was published [72].

This paper presents a reliable procedure for the stability improvement of polymer materials by means of algal extracts. Samples consisting of ethylene–propylene copolymer are modified by various loadings of *Ascophyllum nodosum* extract powder. The protection effect offered by this additive is an example for any other packing material, whose use may be a reliable option for ecological products.

## 2. Materials and Methods

The studied polymeric material, ethylene–propylene copolymer (EPR), is an elastomer produced by ARPECHIM (Pitesti, Romania) as TERPIT C. The pristine rubber has an ethylene/propylene ratio of 3:1. Algal extract from kelp (*Ascophyllum nodosum*) harvested between May and November in Canada was available from the market by Z-Company (Eindhoven, the Netherlands).

The polymer samples were prepared by the dissolution of elastomer in chloroform, whose evaporation at room temperature leaves unchanged the polymer film. After the filtration of this primary solution, aliquots of 10 mL were transferred into three other separate glass flasks, where appropriate amounts of algal powder were put in for the preparation of three other second set solutions containing 0.5, 1, and 2 phr of additive. These last solutions were the sample sources from which 100 mL of liquid was poured into previously weighted aluminum round pans. After gentle drying on the table at room temperature, thin films were obtained, whose weights are placed around 3 mg.

The γ-exposure of probes was accomplished in air at room temperature in a specialized machine, Ob Servo Sanguis type irradiation equipment (Budapest, Hungary) provided with ^60^Co source at four total doses of 0, 25, 50, and 100 kGy by permanent rotation of the processing can. Gamma irradiation of polymer samples was carried out at a dose rate of 0.5 kGy h^−1^, which is a convenient value for the attendance of oxidative degradation. Occasionally, for the study of radiation effects on algal extract, two doses of 12.5 kGy and 75 kGy were also applied. Both control and modified samples were investigated immediately after the end of each irradiation, avoiding any structural modification due to the decay of short-life radicals.

Chemiluminescence (CL) measurements are considered the most appropriate analytical procedure through which the induced effects of γ-radiolysis, a convenient accelerated procedure, may be pertinently controlled. The CL determinations were carried out with LUMIPOL 3 (Institute of Polymers, Slovak Academy of Sciences, Bratislava, Slovakia), when the evaluation of degradation could be performed with a low temperature error (±0.5 °C). This extremely sensitive method for the investigation [73] of structural modifications occurring in the studied polymer matrices is based on photon emission through the de-excitation of carbonyl compounds when they are formed by the reactions of free radicals with molecular oxygen [74]. For the isothermal CL measurements, three values of temperatures (160 °C, 170 °C, and 180 °C) were selected, as they were appropriate investigation conditions for their convenient oxidation rates. For nonisothermal CL assay a suitable heating rate, 10 °C min^−1^, was preferred. A similar measurement parameter (10 °C min^−1^) was applied when the stability evaluation was conducted by heating specimens immersed in water and physiological serum at 80 °C for 5 h and 10 h. The confidence of CL values is very high because the average differences between the analogous values for emission intensities are ±50 Hz. This allows for a very low measurement error of less than 10^−2^%.

## 3. Results

Evaluation of the stabilization efficiency of the studied material, the solid extract of brown microalgae *Ascophyllum nodosum*, is highly related its stability. During γ-radiolysis of the powder, the degradation of active components in algal extract takes place through depolymerization [75] that leads to a gelation of aqueous solutions or intramolecular fragmentation due to differences in the energies of the bonds. Unfortunately, the structural modifications induced by accelerated degradation induced by γ-rays were not reported and, consequently, any comparison is not possible. However, in Figure 2, some fundamental features related to modifications in the chemistry of kelp powder may be revealed:−The thermal or radiation degradation occurring in polymer matrix progresses through the fragmentation of molecules. These scissions are preferentially produced in EPR in polypropylene units, because the bonds of tertiary carbon atoms have a lower energy. The radicals may be subjected to oxidation through their reactions with diffused oxygen or they participate in crosslinking when the recombination reforms the polymer structure and increases the stability of the material. The presence of any antioxidant turns the evolution of the radicals into stabilization. Though there is a difference between thermal degradation and radioinduced degradation based on the local concentration of radicals, both processes have similar mechanisms based on the propagation stage as the median step [76]. In the presence of antioxidants, the competition between oxidation as degradation process and crosslinking as the improvement route is clearly gained by the latter, because the decay of radicals is achieved by their recombination. Thus, the stabilized material is characterized by the relevant contribution of antioxidant to the delay of oxidation by the protection of radicals against their conversion into oxygenated structures.−At low irradiation dose (12.5 kGy), early degradation starts, which is proved by the intensity peak at 75 °C. This photon emission would be caused by the molecular fragmentation into large moieties. This maximum will be never noticed in the other CL spectra obtained on the algal extract subjected to intensive damaging action of γ-radiation. This assumption is valid as sustained by detailed investigations of radiation effects on polysaccharides [77,78]. Additionally, this dose range is suitable for the preparation of hydrogels starting from polysaccharides, when the crosslinking of processed material is certainly reached by the further reactions of these intermediates [79].−The presence of permanent maximum, which appeared at 165 °C, indicates the vulnerability of components upon the energy transfer occurring on the radiation tracks. The evolution of radiochemical degradation is described by the increase in the values of emission intensities, which are the consequence of scission in the structure of monosaccharide units. This process always occurs in all similar molecules, whose radiation stabilities depend on molecular construction [80].−The common joint point may define a certain limit in the reactivity of fragments with respect to oxidation when degradation is guided at high temperatures.

Analysis of families of CL isothermal spectra (Figure 3) demonstrates the capacity of algal extracts from *Ascophyllum nodosum* to extend service life of the studied elastomeric material. The increase of additive concentration in the studied compositions leads to an extension of degradation periods, where the propagation step becomes longer (Figure 3). If the loadings of *Ascophyllum nodosum* extract are greater than 1 phr, the oxidation rates for these samples are extremely low at 160 °C. This means that the operation of items manufactured as modified EPR products is safe for a very long period. The delay of oxidation is indubitably conditioned by the scavenging activities of active components on protection material, especially alginic acid. The long degradation periods are demonstrative proof of improved stability when packaging materials are left in inappropriate conditions such as sunlight and accidental heating. The contribution of algal additive is relevant when its concentration is 2 phr and the degradation temperature does not exceed 100 °C. This situation scarcely occurs. Thus, the presence of algal extract is a guaranteed formulation for long-life products, which operate under ageing conditions.

In considering the possibilities offered by the presence of antioxidants in the formulation of polymer materials, it is easy to identify that the algal extracts are active materials and that their contribution to the extension of durability is related to protection against oxidation by the scavenging of radicals [81]. Relevant components in extracts of *Ascophyllum nodosum* [82] are possible antioxidants with different stabilization efficiencies. Oxidation as a harmful process can be avoided by the addition of appropriate compounds, such as seaweed powder extracts [83]. Their scavenging activities are influenced by the complex composition through which a synergetic effect occurs. This assessment offers the possibility to connect their efficiencies with the functional performances of materials where they are active. Material lifespan is influenced by the initiation of inactivation of radical reactions by seaweed extracts; long-term stability is guaranteed by the antioxidant capacity in natural environments [84].

The γ-irradiation of EPR samples produces differences between pristine material and the polymer improved by algal extract (Figure 4). The degradation periods for EPR in the presence of algal powder are much longer and the emission intensities for the three formulations are significantly reduced. This benefit is evidence of the contribution of oxidation protection through which polymer support may gain an extended durability.

The nonisothermal spectra that depict the development of oxidation in the EPR samples (Figure 4) reveal the stabilization effects of algal extract on the host polymer material under γ-irradiation as well as in the pristine samples. The anti-ageing efficiency that increases as the concentration of additive is increased illustrates the significance of this kind of material in the formulation of packaging materials subjected to accelerating damage in different thermal hazardous environments.

Radiolysis action modifies the oxidation states of materials to quicker degradation when the initiation of oxidation appears earlier under higher exposure of γ-dose. As stated earlier [85], polysaccharides extracts from seaweed are intelligent protectors on packaging materials, because they induce the extension of stability due to the descendants, namely 1,4-β-D-mannuronic acid and 1,4-α-L-guluronic acid, whose sequential distribution may influence the amplitudes of activities.

The modification of temperature values for the start of oxidation is correlated with the exposure dose because the molecular breaking generates not only polymeric forms of descendants (polymannuronic acid and polyguluronic acid), but also monomers which are able to continue the protection action by means of mobile protons of hydroxyl moieties. Because alginic acid is a hydrophobic polysaccharide and the antioxidant activity is determined by the active hydroxyls [86], the width of application ranges may be enlarged by the derivation under the action of γ-radiation according with the results presented in Figure 5.

The inclusion of alginic acid or alginates in the composition of packaging polymer materials allows the implementation of radiation sterilization without affecting the material’s qualities. The radiation processing of these compounds causes structural rearrangements, which may offer a certain level of protection [87,88].

The degradation of EPR/*Ascophyllum nodosum* powder samples in water and physiological serum (Figure 6) is subjected to the effects of interference with water. The water dipoles act as hydrolysis agent [89] as well as attack agent in thermal [90] and radiation [91] environments. The degradation rates present greater values for the modified polymer, while the neat EPR resist for longer periods in aqueous solutions. At the same time, the CL spectra recorded on EPR/algal extract samples does not differ which suggests that the concentration of stabilizer does not significantly affect progress in the oxidation of polymer substrate.

## 4. Discussion

The stability of materials, a fundamental feature in service periods, is influenced by the presence of a compound acting as a degradation protector. As it was previously demonstrated, inorganic structures are able to play the role of anti-ageing factor [92,93,94]. Interest in the extension of the life of materials using ecological products has focused on algal extracts, whose efficiency is one of the main factors in the evaluation of its utility [95]. The significant role of additives for protection against oxidation is exemplified when the local concentration of free radicals reaches a certain level from which the oxidation rate exceeds the accumulation rate [95].

The radiolysis effects on the studied extract of *Ascophyllum nodosum* are characterized by the increase in the oxidation state of processed material due to the scission of molecules. If the molecules of polysaccharide are split as the first degradation step, this process generates radicals available for oxidation [96]. Thus, the first peak at 80 °C appears at low γ-exposure dose (Figure 2). Once the fragments are generated, the rate of their oxidation becomes the dominant process, when the accumulation of peroxyl radicals advances sharply (Figure 2). At the dose exceeding 50 kGy, both processes take place simultaneously. The overlapping of these maxima on the curve recorded for the sample irradiated at 75 kGy is proof for the development of these processes at the same time. However, the descendants of molecular splitting (mannuronic acid and guluronic acid) present antioxidant features which were already demonstrated [97].

The progress of oxidation in the polymer bulk is conditioned by the activity of the used additive, which acts as any hindered phenolic structures [4] (Figure 7). The Bolland–Gee mechanism [66] that describes the oxidation of the elastomer sample explains the sigmoidal shape of isothermal CL spectra. The protection action of added algal extraction is present at the propagation stage of degradation, when the free radicals generated by scissions are tightly scavenged by the alginic acid or its descendants. The supplementary activities of mannuronic and guluronic acids are also related to the extended stabilization periods when the prolongation of the propagation stage of oxidative degradation over 400 min is at a striking discrepancy with the behavior of pristine polymer.

The sustained antioxidant activity of neat and irradiated extract of *Ascophyllum nodosum* is also beneficial for stability when materials are stored, or they are used for food preservation. The predicted stability from the contribution of *Ascophyllum nodosum* powder corresponds to the activity of hydroxyls existing in the structure of the additive [98,99]. Antioxidant responses to harmful conditions are related to the characteristics of polyphenols, whose mobile protons are substituted by free radicals (Figure 7) [100].

The nonisothermal chemiluminescence assays on the antioxidant contributions of the solid extract from *Ascophyllum nodosum* reveal its contribution as the inhibitor of oxidation in the solid polymer probes subjected to the destructive action of γ-radiation in air (Figure 5). The effects are more visible on the high temperature range over 150 °C, when thermal movement is accelerated and the probability of jointing radicals onto active phenol hydroxyls is higher. The best results were obtained when the additive concentration was 2 phr. The existence of a prominent peak at 220 °C demonstrates the formation of structured peroxyls after a longer degradation period, and their decay takes place much later. The availability of active descendant fragments is proved by the increase in their abundance due to the radiation’s damaging effects on macromolecular components [64]. The promising results obtained in the presence of algal extract compose the protective picture by which phenolic structures are involved via the formation of methoxy moieties after the scavenging free radicals.

Unfortunately, the dissimilar effect of algal extracts is noticed in aqueous environments, distilled water, and physiological serum. The nonisothermal CL curves recorded on the EPR specimens modified with *Ascophyllum nodosum* powder are placed under the curve for pristine material. This feature would be ascribed to hydrolysis of the additive, which diminishes protection capacity. However, the lack of contrary effect on the low temperature range up to 100 °C is a positive behavior through which the active configurations are available for moderate stabilization.

## 5. Conclusions

This paper presents relevant results through which the application of ethylene–propylene elastomer may be used as a safe material for the preservation and handling of food. Stability investigation through isothermal and nonisothermal chemiluminescence proves the good protective activity of *Ascophyllum nodosum* powder. The values of kinetic parameters, namely oxidation induction times and onset oxidation temperature, are appropriate evidence when the polymer is subjected to accelerated degradation by γ-irradiation. While the increase of induction period is revealed, the increase in the temperature when the effective oxidation starts indicates the significant anti-ageing activity of the additive. The propagation of oxidative degradation is delayed by the replacement of protons existing in the alginic acid, the main active component of *Ascophyllum nodosum* extract. The stability testing achieved by radiation processing provides strong proof for the useful contribution of algal extracts to the extension of material durability in wide areas of application. An interesting opportunity for the use of *Ascophyllum nodosum* powder in the formulation of polymer materials is its addition to packaging sheets or beverage bottles.

The extension of application ranges for algal extracts opens numerous doors through which nature offers suitable solutions for a healthy life and convenient and economical versions of product manufacturing.

## Figures and Tables

**Figure 1 polymers-15-00977-f001:**
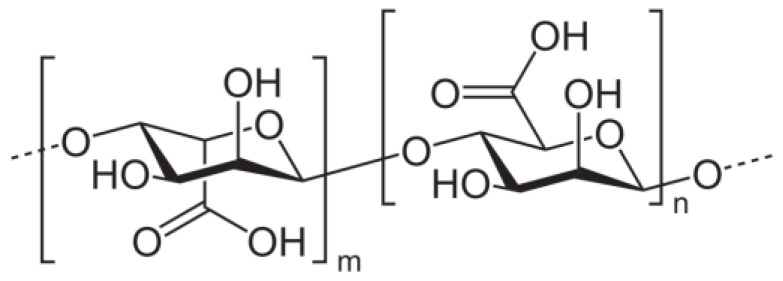
The molecular structure of alginic acid.

**Figure 2 polymers-15-00977-f002:**
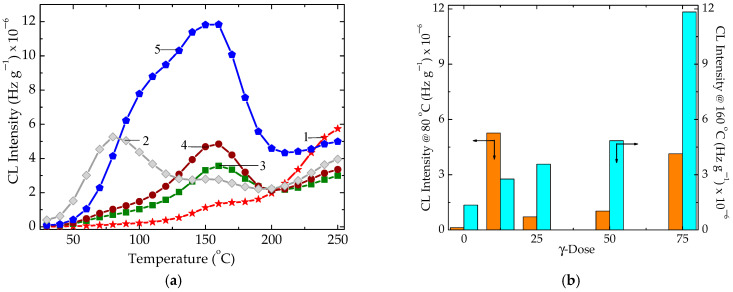
(**a**) Nonisothermal CL spectra obtained on the kelp powder γ-irradiated at several doses. Heating rate: 10 °C min^−1^. Exposure doses: (1) 0 kGy; (2) 12.5 kGy; (3) 25 kGy; (4) 50 kGy; (5) 75 kGy. (**b**) Histogram of oxidation development by means of the two intensity peaks that appeared in CL nonisothermal spectra.

**Figure 3 polymers-15-00977-f003:**
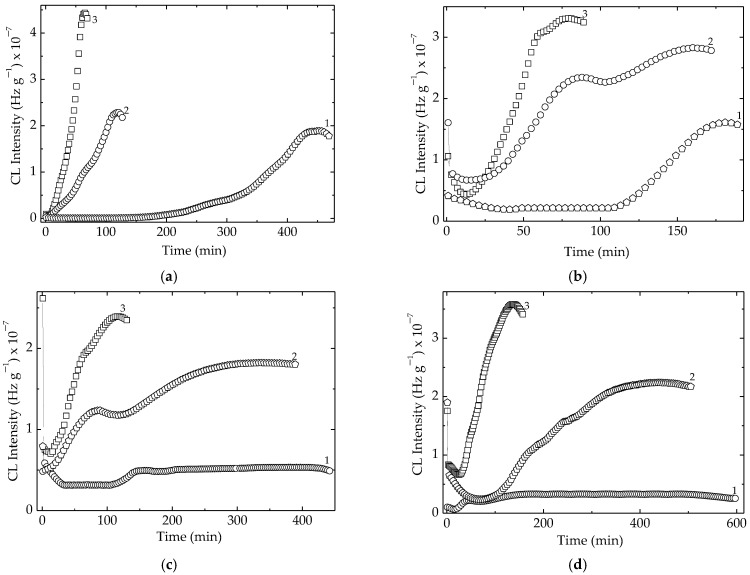
The isothermal CL spectra recorded on nonirradiated EPR loaded with different kelp powder amounts. (**a**) control; (**b**) 0.5 phr; (**c**) 1 phr; (**d**) 2 phr. Testing temperatures: (1) 160 °C; (2) 170 °C; (3) 180 °C.

**Figure 4 polymers-15-00977-f004:**
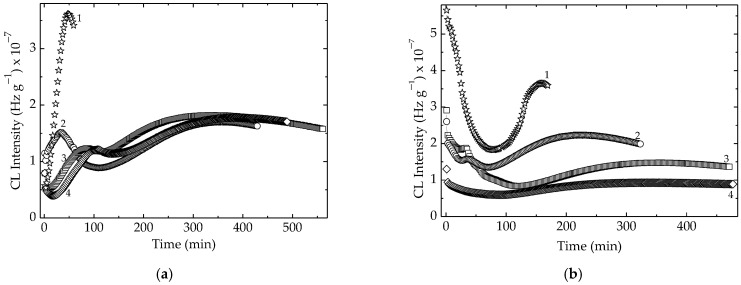
The isothermal CL spectra recorded on EPR specimens with various concentrations of solid *Ascophyllum nodosum* extract. Exposure doses: (**a**) 25 kGy; (**b**) 100 kGy. (1) pristine polymer; (2) polymer + 0.5 phr extract; (3) polymer + 1 phr extract; (4) polymer + 2 phr extract. Testing temperature: 170 °C.

**Figure 5 polymers-15-00977-f005:**
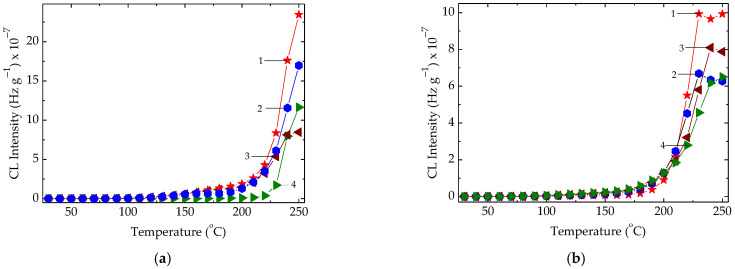
The CL nonisothermal spectra recorded consisting of EPR and *Ascophyllum nodosum* powder. Heating rate: 10 °C min^−1^. Dose: (**a**) 0 kGy; (**b**) 25 kGy; (**c**) 50 kGy; (**d**) 100 kGy. Composition: (1) neat polymer; (2) EPR + 0.5 phr powder; (3) EPR + 1 phr powder; (4) EPR + 2 phr powder.

**Figure 6 polymers-15-00977-f006:**
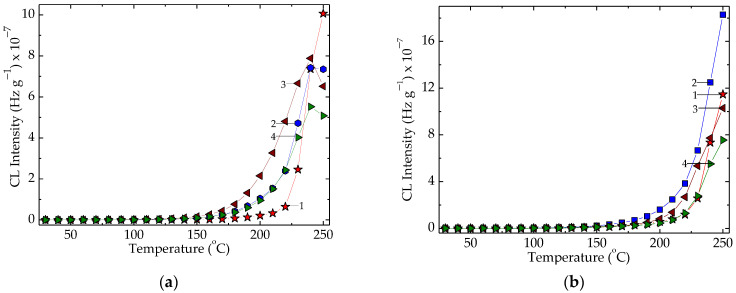
The nonisothermal CL spectra recorded on the EPR/*Ascophyllum nodosum* samples immersed in aqueous environments subjected to heat treatment at 80 °C (**a**–**d**) and irradiation at 25 kGy (**e**,**f**). Heating rate: 10 °C min^−1^. (**a**) water for 5 h; (**b**) physiological serum for 5 h; (**c**) water for 10 h; (**d**) physiological serum for 10 h; (**e**) water; (**f**) physiological serum. (1) control EPR; (2) EPR + kelp 0.5 phr; (3) EPR + kelp 1 phr; (4) EPR + kelp 2 phr.

**Figure 7 polymers-15-00977-f007:**
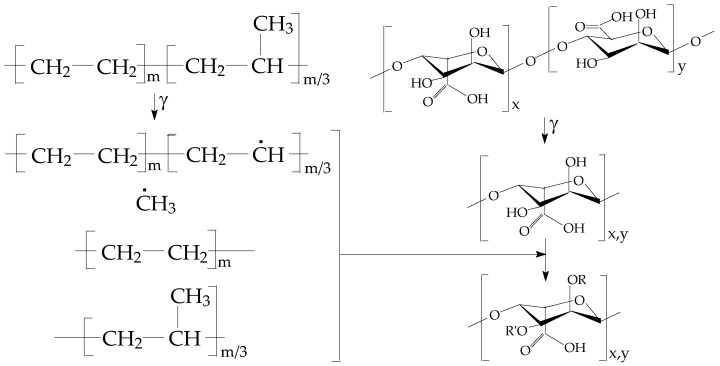
The proposed mechanistic scheme for the stabilization activity of alginic acid.

## Data Availability

The data presented in this study are available on request from the corresponding author.

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
