# Peer review of "Stabilization Activity of Kelp Extract in Ethylene–Propylene Rubber as Safe Packaging Material"

_polymers, 2023, doi:10.3390/polym15040977_

Round 1

Reviewer 1 Report

The reviewed article presents the results of a study on the stabilising effect of KELP extract on ethylene-propylene copolymer. The developed material can be used as a packaging material.
In my opinion, the results presented are promising, especially with regard to the use of natural substances in modifying the properties of polymeric materials.
Nevertheless, I have a question regarding the amount of algal extract used: why was a maximum of 2phr of this ingredient used for the modification? Has the detailed composition of the extract used been investigated?  
Please check the captions under Fig.2. in my opinion there are some inaccuracies in the description there.
The manuscript has been reliably prepared by the authors and with minor corrections may be considered for publication in the journal Polymers

Author Response

Thank you for all comments and suggestions. They are appropriate.

Reviewer 2 Report

The work is nicely written. Below more minor comments:

L 103 how is the 10°C ramp appropriate. Can a reference be given?

L 141 depolymerization. Please the mechanism. Later on some molecular are mentioned (L 200) but this can be put in a more direct manner.

L 147 a general reader: low dose, early degradation (one does not expect that as in high even earlier).

L 165. Please give a general hint on the formation of radicals or intermediates. Are overall activation energies know. If not mention that for these polymers one thus not have the same knowledge level as for e.g. polyacrylics (one can cite Polymers 2020 12, 1667)

Caption 3: avoid the use of only abbreviations.

General comment: the (general) reader needs to be more convinced of why irradiation is the trigger for degradation as opposed to water contact of temperature as such.

Author Response

(The authors gave the same response as above.)
